# Risk Factors for Feeding and Swallowing Disorders in Very Low Birth Weight Infants in Their Second Year of Life

**DOI:** 10.3390/medicina58111536

**Published:** 2022-10-27

**Authors:** Nuša Slana, Irena Hočevar-Boltežar, Lilijana Kornhauser-Cerar

**Affiliations:** 1Department for (Re)Habilitation of Children, University Rehabilitation Institute Republic of Slovenia Soča, SI-1000 Ljubljana, Slovenia; 2Department of Special Education and Rehabilitation, Faculty of Education, University of Ljubljana, SI-1000 Ljubljana, Slovenia; 3Department of Otorhinolaryngology and Cervicofacial Surgery, University Medical Center Ljubljana, SI-1525 Ljubljana, Slovenia; 4Department of Otorhinolaryngology and Cervicofacial Surgery, Faculty of Medicine, University of Ljubljana, SI-1000 Ljubljana, Slovenia; 5Neonatal Intensive Care Unit, Department for Obstetrics and Gynaecology, Division for Perinatology, University Medical Center Ljubljana, SI-1525 Ljubljana, Slovenia

**Keywords:** feeding and swallowing disorder, growth, prematurity, speech development, very low birth weight

## Abstract

*Background and Objectives*: This study aimed to identify the prevalence of feeding and swallowing disorders (FSD) in very low birth weight (VLBW, 1500 g or less) infants in the first two years after discharge from the maternity hospital, their possible risk factors, and the consequences of them. *Materials and Methods*: A total of 117 preterm children with VLBW born between 2013 and 2015 were included. The data concerning possible FSD after discharge from the hospital were obtained through accessible medical documentation for the child and a short parental questionnaire. *Results*: FSD was reported in 32 (27.4%) infants following discharge from the hospital but in only five children (4.3%) at a mean age of four years. Four variables (birth gestational age less than 28 weeks, birth weight equal to or less than 1000 g, birth length below 33 cm, and start of oral feeding after the 34th gestational week) were identified as risk factors for FSD after discharge. However, only birth length remained a significant predictor after being included in a binary logistic regression model (*p* = 0.000). Abnormal oral sensitivity and a decrease in weight to under the 10th percentile were significantly more common in the FSD group at follow-up visits at the age of about 2 years. *Conclusions*: FSD was still present in more than one-quarter of VLBW infants after discharge from the maternity hospital but mostly disappeared within four years. A birth gestational age under 28 weeks, weight up to 1000 g, the late beginning of per oral feeding, and a birth length below 33 cm were determined to be significant predictive factors for FSD. Having a birth length below 33 cm was associated with an almost 6.5-fold increase in the odds of having persistent FSD after discharge from the hospital. FSD in the first years of life may have an impact on the child’s further growth and development.

## 1. Introduction

Ultrasound studies have shown that non-nutritive sucking and swallowing appear in most fetuses by 15 weeks of gestation [1]. Consistent swallowing is noticed by 22–24 gestation weeks [2]. The central nervous control and efficiency of the peripheral organs involved in breathing, sucking, and swallowing are not sufficiently developed for oral feeding before the age of 33−34 gestational weeks [3,4]. Because premature birth abruptly interrupts the child’s development in the uterus, preterm infants often have feeding and swallowing disorders (FSD), with a prevalence ranging from 40−70% [5,6]. The incidence of dysphagia in preterm infants is increasing, due in part to the improved survival rates of premature newborns, especially of those with complex medical conditions [7]. Medical complications that are related to preterm birth and/or that can adversely affect swallowing and feeding are respiratory distress syndrome, bronchopulmonary dysplasia, cardiac problems resulting in apnea and bradycardia, necrotizing enterocolitis, and neurological disorders [7,8,9,10,11]. However, the primary cause for swallowing disorders in preterm infants is probably their neurodevelopmental immaturity [12].

The main condition for discharge from the maternity hospital is successful feeding and sufficient weight gain of the newborn. The delay in acquiring feeding skills is present in premature newborns who are less mature at birth (at or under 28 weeks gestation), have an extremely low birth weight (1000 g or less), and have complex respiratory, neurologic, and gastrointestinal comorbidities [13,14,15]. Some studies have reported persistent feeding and swallowing problems in preterm infants after discharge from the hospital. Researchers estimate that the incidence of persistent feeding problems in preterm newborns ranges from 19 to 80% [16,17]. The feeding difficulties may remain a problem for former preterm neonates well into the toddler years and even later [16]. Their parents report a prolonged feeding time, feeding refusal, poor acceptance of textured foods, coughing/choking during swallowing, oral-motor problems, poor chewing, and the need for professional help in feeding in the first few years of the child’s life [18]. These are reasons that a considerable number of mothers often feel discomfort during feeding their preterm babies at least some months after leaving the maternity hospital [19].

Some researchers have tried to detect the factors predicting persistent swallowing problems after discharge from the hospital. Hoogewerf et al. followed a group of neonates (premature and at-term) who needed intensive care soon after birth. The prevalence of feeding problems was high in this population (20.4%), but each gestational age group showed a similar percentage of feeding problems of around 20%. Prolonged tube feeding was the most important risk factor in premature newborns. Being born small for gestational age was the most prevalent risk factor for children born after 32 weeks of gestational age [20].

Meta-analysis of 22 studies published in English between 2000 and 2020, including 4381 children (six studies including a population of more than 100 children), showed that FSD is even more common in premature-born children in the first four years of their lives. The overall prevalence of FSD was 42%, but the heterogeneity across the studies was significant. The prevalence in children born extremely prematurely (<28 weeks of gestational age at birth) was 46%. For children born at a gestational age of 28–32 weeks and 32–37 weeks, the prevalence of FSD was 42% and 38%, respectively. However, the authors concluded that the degree of prematurity did not influence the prevalence of FSD. There were also some longitudinal studies included in the meta-analysis. The prevalence of FSD in premature-born children was 43% at the age of 0–5 months, 38% at the age of 6–11 months, 33% at the age of 12–23 months, and 33% at the age of 24–48 months [21]. 

According to the National Perinatal Information System of Slovenia, which serves as Slovenia’s birth registry, preterm children represent 6−7% of all children born in Slovenia; fewer than 1.5% have a birth weight of 1500 g or less (very low birth weight - VLBW infants) [22]. There have not been many studies on FSD after discharge from the maternity hospital in VLBW infants. The aim of this study was to identify the prevalence of FSD in VLBW preterm infants in the first two years of their lives, the factors which can predict this FSD, and its possible connection to the child’s development.

## 2. Materials and Methods

The study was carried out in accordance with The Code of Ethics of the World Medical Association (Declaration of Helsinki) and approved by the National Medical Ethics Committee of the Republic of Slovenia (Protocol No. 0120-112/2018/4).

### 2.1. Questionnaire

The questionnaire included the child’s general information and questions concerning the feeding method used after discharge from the maternity hospital (breastfeeding, bottle/cup/syringe feeding with mother’s milk or formula, use of a nasogastric tube or gastrostomy), possible abnormal oral sensitivity, and problems with feeding and swallowing (prolonged feeding time, feeding refusal, problems with acceptance of textured foods and chewing, coughing/choking during swallowing, oral-motor problems) at the time the questionnaire was filled in. 

### 2.2. Medical Documentation

The questionnaire also included a request to access the child’s medical records. If the parents signed the informed consent, data were obtained from the medical documentation of the maternity hospital. 

The great majority of the children were followed up for two years by a pediatrician from the maternity hospital. The data on evident swallowing problems (as stated in the above questionnaire), a lack of general interest in food, and reports on picky eaters were considered as reported FSD. These feeding problems were mentioned in the medical documentation because parents or the pediatrician brought them up as an important issue or problem. Data on the possible assessment methods of these problems (parents’ reports, Pediatric Assessment Tool, Schedule for Oral Motor Assessment, clinical observation, Flexible Endoscopic Evaluation of Swallowing) were not found in the majority of the records, and therefore, they were not included in the analysis. 

The results of the validated Slovenian version [23] of the Denver Developmental Screening test II (DDST II) [24], which was performed at the last follow-up visit in order to detect any possible deviation from normal development, recorded the infant’s weight at the last follow-up, the weight percentile of the infant, and possible documented FSD, which were taken into consideration for the present study. 

### 2.3. Participants

The inclusion criteria for participants were as follows: being born preterm in the Ljubljana Maternity Hospital between 2013 and 2015 and having a birth weight of 1500 g or less (VLBW infants). A questionnaire and a signed informed consent were returned by the parents of 136 children (35%) that met both criteria. Nineteen children were not followed-up by the pediatrician from the maternity hospital and were excluded from the study; as a result, 117 children in total were included. 

### 2.4. Statistical Analyses

Analyses were performed using the SPSS version 22.0 (SPSS Inc., Chicago, IL, USA). After taking the medical documentation into consideration, 32 infants were included in the group of preterm children with documented FSD. A bivariate analysis comparing the newborn’s characteristics, the possible predictive factors for persistent FSD after discharge from the maternity hospital, and possible consequences of persistent FSD on the child’s development was conducted between the two groups of VLBW preterm infants with and without FSD. A t-test, the nonparametric Mann–Whitney test, χ^2^, and Fisher’s exact test were used as appropriate. The statistically significant parameters considered predictors of persistent FSD were included in a binary logistic regression model. Non-significant variables were removed one by one, removing the largest *p*-value first, until all the remaining variables in the model were significant. All tests were considered significant when *p* < 0.05. 

## 3. Results

The sample of the 117 included children consisting of 65 (55.6%) girls and 52 (44.4%) boys. The average age of the participants at the time of the study ranged from 31 to 69 months, with a mean age of 48.4 months (standard deviation—SD 10.8 months). 

The data from the medical documentation until discharge, as well as from the follow-up visits at the Maternity Hospital Ljubljana are presented in Table 1. The child’s associated comorbidities possibly influencing feeding and swallowing included bronchopulmonary dysplasia (4 newborns), a cerebral lesion detected by ultrasound (29 newborns, intracranial hemorrhage in 22 newborns), patent ductus arteriosus (3 newborns), severe gastrointestinal disorders (4 newborns), and cleft lip and palate (1 newborn). 

All preterm children were fed by a nasogastric or orogastric tube from birth onwards; however, complementary parenteral feeding was necessary for 113 (96.6%) until the infants tolerated a sufficient amount of milk by the tube. Tube feeding continued until sucking, swallowing, and breathing were coordinated enough for sufficient per oral intake. The children were discharged from the maternity hospital when their health status was stable, they were completely fed by mouth, and were gaining weight. 

All the included children were followed by a pediatrician at the Ljubljana Maternity Hospital until the age of 19–32 months (mean 24.07 months, SD 1.87 months). During their follow-up visits, 20 children manifested swallowing disorders (two of them requiring a nasogastric tube for feeding for between 2 and 4 months, respectively, because of insufficient weight gain), one child was a picky eater, and 11 children showed little interest in food in general with prolonged meals and the refusal of certain kinds of food. 

In the questionnaire, parents of 19 children (16.2%) reported that their children had abnormal oral sensitivity. Twenty-three children (19.7%) were breastfed with or without supplements, all others were fed by bottle, syringe, or cup. No child needed a gastrostomy for feeding. Only five (3.7%) of the VLBW preterm children were reported by their parents to have persistent difficulties with feeding and swallowing at the time the questionnaire was filled out at a mean age of 48.4 months (SD 10.6 months). All five children had FSD after their discharge from the maternity hospital and at the age of two years at their last follow-up visit at the maternity hospital.

The bivariate analysis of birth variables and the possible predictive factors for FSD comparing the VLBW preterm infants with FSD after discharge from the maternity hospital versus those preterm VLBW infants without such disorders showed statistically significant differences for four parameters: gestational age at birth (especially less than 28 gestational weeks), birth weight (especially equal or less than 1000 g), birth length (especially below 33 cm), and the beginning of oral feeding after a gestational age of 34 weeks (Table 2). These parameters from the bivariate analysis were included in a binary logistic regression model. When the four above-mentioned parameters were analyzed together, no significant effect for any of the variables was noticed. When the parameter with the largest *p* (birth weight equal to or less than 1000 g, *p* = 0.631) was removed from the model, only birth length became significant (*p* = 0.006). After removing the next insignificant parameter from the model (the beginning of oral feeding after the gestational age of 34 weeks; *p* = 0.256), only birth length below 33 cm remained significant (*p* = 0.003). In the next step, the last insignificant parameter, of a gestation age less than 28 weeks, was removed (*p* = 0.166). A birth length below 33 cm remained a statistically significant factor (odds ratio 6.481 [95% CI 2.426, 17.318], *p* = 0.000). Therefore, having a birth length smaller than 33 cm was associated with an almost 6.5-fold increase in the odds of having persistent FSD after discharge from the hospital. On the other hand, having the birth length below the 10th centile (“short for gestational age”) was not a risk factor for FSD, and there were no significant differences between the two groups with respect to ponderal index (a marker to assess body proportionality with the aim to distinguish symmetric from asymmetric growth restriction at birth, calculated by the formula: weight (g)/(length in cm)^3^ × 100).

At the last follow-up visit to the maternity hospital, infant weight was above or equal to the 10th percentile in 80 children, remained below the 10th percentile in 9 children, and fell from above the 10th percentile to below it in 28 children. 

DDST II detected five children with a global developmental disorder and 18 children with a language development disorder. FSD was present in all five children with a global developmental disorder and in nine children (50%) with only language development disorders.

When the possible consequences of persistent FSD on a child’s growth and language development were compared between the group of children with FSD and the group without FSD, a significant difference was found in three parameters (oral sensitivity, weight percentile at the last follow-up visit, and weight drop from above the 10th percentile at birth to below the 10th percentile at the last follow-up visit). The prevalence of language development disorders according to DDST II was close to statistical significance (Table 3). FSD was persistent after discharge from the hospital in 19/28 (67.9%) children with a weight drop to below the 10th percentile at the last follow-up visit at the age of about two years.

Thirteen children with FSD were included in speech and language therapy, namely nine children only for language and articulation problems and five children for FSD. These five children were referred to a tertiary hospital that specialized in the rehabilitation of FSD at a mean age of 18 months. Four children were included in therapy for sensory-motor feeding problems, while the parents of one child were counseled about the child’s delayed feeding skills. 

## 4. Discussion

Due to an immature upper aerodigestive tract and immaturity in its neural control, almost all of the preterm new-borns (especially those with a birth weight under 1500 g) initially had to be fed parenterally and/or by tube [25], which was also the case in our study. Only when feeding was satisfactory enough to allow for a proper increase in the infant’s weight was the child discharged from the maternity hospital and sent home. The Ljubljana Maternity Hospital is a tertiary-level perinatal centre in which more than 80% of Slovene VLBW infants are delivered [26,27]. The Neonatal Intensive Care Unit (NICU) of the Ljubljana Maternity Hospital is properly equipped and has staff experienced in the demanding care of all its infants; since 2008, the NICU has been a member of the Vermont Oxford Network (VON) [28]. This may be the reason that the percentage of VLBW children with persistent FSD was lower than reported in the literature [21]. After discharge, FSD was still reported in follow-up medical documentation in 32 children (27.4%) up to a mean age of about two years. We decided to include the children who did not show a general interest in food and the picky eaters in the group with FSD. Selectivity in food and a decreased interest in food may be consequences of previous swallowing disorders (e.g., frequent vomiting or coughing during bottle feeding can cause a child’s adverse reaction to taking food). These feeding problems were mentioned in the medical documentation because parents or the pediatrician brought them up as an important issue or problem. Nevertheless, parents of only 5 children (4.3%) reported FSD at the time of the study, when their children were about four years old. Therefore, our findings are better than those reported in the meta-analysis, which included 22 studies about the problematic feeding of prematurely born children under 4 years of age whose data showed a prevalence of FSD in 33% and 42%, in age groups 12–23 months and 24–48 months, respectively [21]. 

In most of the studies and reviews, low gestational age at birth, low birth weight, and accompanying morbidities associated with prematurity are cited as factors affecting feeding and swallowing in infants [3,29,30]. The results of our study confirmed that gestational age at birth (especially below 28 weeks) and birth weight (especially equal to or less than 1000 g) are connected to persistent FSD. We did not confirm pre- or perinatal brain damage to be one of the risk factors for persistent FSD after discharge from the hospital. The other accompanying morbidities were very rare among the included neonates (affecting 1–4 children out of the 117 included); therefore, the statistical analysis was not meaningful. However, birth length and the beginning of oral feeding after the age of 34 gestational weeks were shown to be factors influencing the FSD in the group of premature infants with VLBW. The statistically important effect of low birth weight, low birth length, and low gestational age at birth were expected as all variables are interconnected. It is also known that children with a birth weight below 1000 g are at much greater risk of later health and developmental issues [31]. 

The results of our study showed that birth length appeared to be a significant factor influencing FSD in the group of premature infants with VLBW. Although some researchers have cautioned against the use of birth length as a significant infant characteristic because of difficulties in straightening a preterm newborn and thus risking the underestimation of the length [32], this type of mismeasuring constitutes a “non-differential moving effect” that nullifies itself since all the included children were measured in the same way. The other limitation would be that the infants’ lengths at birth were not adjusted for their parents’ heights. A database with more detailed parental height information would be helpful to adjust the newborn’s length according to its genetic influence on it. The evidence for the independent effect of birth length on a child’s health is limited compared to the numerous studies on birth weight or head circumference. However, several studies suggest that children in utero who failed to obtain a proper length relative to their weight seemed to be more vulnerable compared to proportionally small, average-size, or large infants. These infants had higher perinatal mortality [33], higher hospitalization rates from 6 to 18 months [34], and an increased risk of bronchopulmonary dysplasia [35] and obesity in early adolescence [36]. As rapid growth in length occurs early in gestation, while the fetus gains weight more rapidly later in gestation, a disproportion between length and weight at birth may be a sign of placental dysfunction or other disorders affecting the fetus occurring at different periods and may also play a role in subsequent health problems in the child, such as FSD.

As the mean birth length of the children with persistent FSD was about 33 cm, we supposed that a birth length below this value would be significantly related to an increased risk for FSD after discharge from the hospital. As a matter of fact, the results of the binary logistic regression analysis showed that newborns with a birth length below 33 cm had 6.5-fold greater odds for FSD than newborns with a longer birth length. Moreover, a birth length below 33 cm remained the only statistically significant parameter when all four parameters showing significance in the univariate analysis were analyzed together.

The central nervous control and the efficiency of the upper respiratory and digestive tracts are sufficiently developed for oral feeding after the age of 34 gestational weeks in the case of a normal child’s development [4]. We hypothesized that the beginning of oral feeding after that time as a milestone in the newborn’s maturation when the coordination between breathing, sucking and swallowing are normally expected is a sign of abnormal neuromuscular control of these three functions of the upper aerodigestive system. This may be due to the disordered development of the control system, either for the same reasons underlying the premature birth or for other reasons.

We also tried to find out whether small-for-date infants (with a birth weight below the 10th percentile) were at greater risk of persistent FSD. According to our results, the weight percentile, length percentile at birth, and ponderal index did not appear to be significant parameters for FSD after their discharge from the hospital. Therefore, we cannot exclude that there may be other factors (e.g., associated comorbidities), not just immaturity and insufficient intrauterine growth, that cause FSD in children. The fact that FSD occurred more frequently in infants who started with oral feeding after a gestational age of 34 weeks speaks in favor of this assumption. Of course, some associated comorbidities in infants from our study can be significant causes for disorders in the coordination of peripheral organs involved in breathing, sucking, and swallowing at the same time (e.g., cleft palate, pulmonary and cardiac issues). Indeed, most reviews note a variety of neuromuscular, cardiorespiratory, gastrointestinal disorders, and structural abnormalities predisposing to FSD in infancy and early childhood [3,37]. On the other hand, when we tried to determine the importance of brain damage for FSD after discharge from the maternity hospital, we did not find a significant correlation.

A statistically significant correlation between FSD and abnormal oral sensitivity was discovered. It is well known that problems with feeding and swallowing are more frequent in children with oral hyper- or hyposensitivity [38]. This was also found in the population of extremely preterm children [39]. Hypersensitivity of the oral cavity is often connected with the use of a feeding tube and no or only limited stimulation of the oral cavity in the first months of an infant’s life. Therefore, the hypersensitivity of the oral cavity is most probably a consequence of, and not the main reason for, swallowing difficulties. This is why we analyzed the disordered sensibility of the oral cavity as a possible consequence of persistent FSD after discharge from the maternity hospital.

At regular follow-up visits, there were significantly more children whose growth did not increase as expected for their age among the VLBW with FSD. There were a considerable number of children who were appropriately developed according to their gestational age at birth, but later their development lagged. In view of the results of the study, it is not possible to say whether the growth retardation is due only to FSD or whether some, as yet undetermined factor influenced the occurrence of FSD and the child’s growth retardation at the same time. On the other hand, Samara et al. [39] reported that extremely preterm children with eating problems were shorter, lighter, and had a lower body mass index even after adjusting for their disabilities, gestational age, birth weight, and feeding problems. Therefore, a persistent FSD after discharge from the hospital represents a real threat to normal body growth and to the development of the child in the future and calls for professional intervention.

A comparison of the infants with and without FSD showed that FSD was present in half of the children in whom a language development delay was registered on screening with the DDST II test at a mean age of two years. It has already been reported that a prior history of feeding and swallowing problems has been found significantly more often in at-term children with language impairments than in children without language problems [40]. However, there are also studies reporting that preterm infants more often show lower cognitive and communicative skills at the age of 2.5 years than children born at term, as well as difficulties in reading comprehension and spelling skills later in childhood [41,42]. The results of our study confirmed the published data showing that about one-fifth of the included VLBW children demonstrated speech development disorders according to a DDST II test performed last at a mean age of about two years. Therefore, we cannot exclude other factors causing FSD besides the prematurity itself from having an impact on speech development. To determine the relationship between FSD and language development problems, a prospective study including a complete team of relevant professionals should be performed from birth to the end of schooling for premature children and at-term children. 

Only 13 children with FSD were included in speech therapy, but only five were actually treated for FSD. The reason for such a small proportion of children who received professional help could be poor access to FSD rehabilitation for children in our country. There are only several institutions with a limited number of properly trained speech pathologists who deal with this problem.

### Deficiencies of the Study

The best way of studying the prevalence of FSD is to use one of the standardized methods (a questionnaire, clinical assessment, or instrumental examination). Unfortunately, information of this kind about FSD was not available in the medical documentation for the majority of children in our study. This problem is quite common in the literature. In the above-mentioned meta-analysis, many of the included studies did not mention the method of assessing problematic feeding in prematurely born infants [21].

Another limitation of our study is the relatively small number of children included (35% of all invited). A longitudinal study would have given more accurate findings about the effect of FSD on the child’s growth and the development of speech and language. 

Despite these limitations, the findings of this study have implications for professionals and parents of preterm children. The study gives insight into the presence of FSD among Slovenian VLBW preterm children, the factors that may predict these disorders, and the long-term consequences they have on preterm children. The effect of FSD on the child’s growth and development is very important information. Additional studies with a higher number of infants included are still needed, however. 

## 5. Conclusions

The results of our study show that more than one-fourth (27.4%) of VLBW infants still had FSD following their discharge from the maternity hospital, especially in cases of birth weight equal to or below 1000 g, gestational age of fewer than 28 weeks at birth, and the beginning of oral feeding after the age of 34 gestational weeks. However, a birth length below 33 cm remained the only significant variable after modelling the significant data in the logistic regression program. This is a novel observation in the literature and needs further research. 

FSD has been noticed in 67.9% of children in whom the weight percentile decreased from normal at birth to below the 10th percentile at the age of about two years, and in 50% of preterm children with speech development disorders, according to DDST II. To ensure the best possible outcome for VLBW infants, an interdisciplinary team of properly qualified professionals should care for infants from birth onward in order to prevent the problems before they occur and/or to treat problems that are already present. 

An encouraging result of the study was that most of the FSD disappeared in a few years. At a mean age of about four years, FSD was present in only 4.3% of the included children.

## Figures and Tables

**Table 1 medicina-58-01536-t001:** Descriptive statistics for data from the medical documentation of the Ljubljana Maternity Hospital (before discharge and at follow-up visits) (*n* = 117).

Variable	Mean/Standard Deviation/Range or *n* (%)
Multiple pregnancy, *n* (%)	30 (25.6%)
Gestational age at birth, wks	27.79/2.43/22–34
Gestational age at birth less than 28 weeks, *n* (%)	54 (46.2%)
Caesarean delivery, *n* (%)	64 (54.7%)
Birth weight, g	1014.99/264.78/370–1500
Percentile of birth weight	37.12/19.82/1–85
Birth weight equal or less than 1000 g, *n* (%)	57 (48.7%)
Birth length, cm	35.63/3.36/24–42
Percentile of birth length	50.85/27.39/1–97
Ponderal index	2.21/0.33/1.63–3.77
Intubation and respiratory support needed, *n* (%)	49 (41.9%)
Non-invasive ventilation needed, *n* (%)	82 (70.1%)
Oxygen supplementation needed, *n* (%)	88 (75.2%)
Brain lesion detected by ultrasound examination, *n* (%)	29 (24.8%)
Phototherapy for jaundice, *n* (%)	73 (62.4%)
Parenteral feeding needed, *n* (%)	113 (96.6%)
Tube feeding needed, *n* (%)	117 (100%)
Gestational age at start of per oral feeding, wks	34.51/1.18/32–40
Gestational age more than 34 weeks at start of per oral feeding, *n* (%)	46 (39.3%)
Weight at discharge from the maternity hospital, g	2242.59/458.73/1430–4500
FSD documented after discharge from the maternity hospital, *n* (%)	32 (27.35%)
Percentile of weight at last follow-up visit	27.91/23.28/1–90
Decrease in weight percentile from normal at birth to below 10th percentile at the last follow-up visit, *n* (%)	28 (23.9%)
Denver Developmental Screening test II.	
- global developmental disorders, *n* (%)	5 (4.3%)
- gross motor skills disorders, *n* (%)	5 (4.3%)
- language development disorders, *n* (%)	18 (15.4%)

g = grams, cm = centimeters, wks = weeks, FSD = feeding and swallowing disorders.

**Table 2 medicina-58-01536-t002:** Comparison of preterm children by feeding and swallowing disorders (FSD) after discharge from the maternity hospital regarding significant possible risk factors for FSD (*n* = 117).

Variable	VLBW Children with FSD *n* = 32	VLBW Children without FSD *n* = 85	*p*
General birth variables			
Male gender, *n*	14 (43.8%)	38 (44.7%)	0.926
Gestational age at birth, wks, M ± SD	26.59 ± 3.03	28.25 ± 2.00	0.007
Birth weight, g	874.13 ± 289.21	1068.02 ± 235.67	0.001 *
Percentile of birth weight, M ± SD	33.68 ± 19.09	38.42 ± 20.06	0.300 *
Birth length, cm, M ± SD	33.48 ± 3.67	36.65 ± 3.37	0.000
Percentile of birth length, M ± SD	48.31 ± 27.08	52.55 ± 27.48	0.303 *
Multiple pregnancy, *n* (%)	7 (21.9%)	23 (27.1%)	0.567
Caesarean delivery, *n* (%)	17 (53.1%)	47 (55.3%)	0.834
Possible predicting factors for FSD			
Gestational age less than 28 weeks, *n* (%)	21 (65.6%)	33 (38.8%)	0.010
Birth weight equal or less than 1000 g, *n* (%)	23 (71.9%)	34 (40%)	0.002
Birth weight below 10th centile, *n* (%)	8 (25%)	12 (14.1%)	0.352
Birth length below 10th centile, *n* (%)	5 (15.6%)	7 (8.2%)	0.240
Birth length below 33 cm, *n* (%)	14 (43.8%)	9 (10.6%)	0.000
Ponderal index, M ± SD	2.28 ± 0.38	2.17 ± 0.35	0.211
Intubation and respiratory support needed, *n* (%)	18 (56.3%)	31 (36.5%)	0.053
Noninvasive ventilation needed, *n* (%)	26 (81.3%)	56 (65.9%)	0.123
Oxygen supplementation needed, *n* (%)	27 (84.4%)	61 (71.8%)	0.186
Brain lesion detected by ultrasound examination, *n* (%)	9 (28.1%)	20 (23.5%)	0.608
Phototherapy for jaundice, *n* (%)	20 (62.5%)	53 (62.4%)	0.988
Gestational age at start of per oral feeding, wks, M ± SD	35.16 ± 1.65	33.85 ± 3.85	0.064
Gestational age more than 34 weeks at start of per oral feeding, *n* (%)	18 (56.3%)	28 (32.9%)	0.024
Breastfeeding, *n* (%)	3 (9.4%)	20 (23.5%)	0.118 **
Weight at discharge from the maternity hospital, g, M ± SD	2370.31 ± 524.55	2193.81 ± 424.38	0.064

* Mann–Whitney test, ** = Fisher exact test. M ± SD = mean standard deviation, g = grams, cm = centimeters, wks = weeks, FSD = feeding and swallowing disorders.

**Table 3 medicina-58-01536-t003:** Comparison of preterm children by feeding and swallowing disorders (FSD) following their discharge from the maternity hospital and possible consequences of FSD and age at last follow-up visit (*n* = 117).

Variable	VLBW Children with FSD *n* = 32	VLBW Children without FSD *n* = 85	*p*
Age at the last follow-up visit, ms, M ± SD	24.88 ± 2.47	23.76 ± 1.5	0.311
Abnormal oral sensitivity, *n* (%)	9 (28.1%)	10 (11.8%)	0.032
Weight percentile at the last follow-up visit, M ± SD	8.72 ± 13.13	35.28 ± 22.03	0.000
Weight drops from above the 10th percentile at birth to below the 10th percentile at the last follow-up visit, *n* (%)	19 (59.4%)	9 (10.6%)	0.000
Language development disorders (DDST II), *n* (%)	12 (37.5%)	12 (14.1%)	0.05

M = mean, SD = standard deviation, ms = months.

## Data Availability

The data published in this research are available on request from the second author (I.H.B.).

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
