# Peer review of "Risk Factors for Feeding and Swallowing Disorders in Very Low Birth Weight Infants in Their Second Year of Life"

_medicina, 2022, doi:10.3390/medicina58111536_

Round 1

Reviewer 1 Report

In their study the authors aimed to identify the prevalence of feeding and swallowing disorders (FSD) in very low birth weight (VLBW) preterm infants in the first two years of their lives, the factors which can predict these FSD, and the possible connection to the child’s development.

Although they did not use the standardized methods (questionnaire, clinical assessment, instrumental examination) and they have small number of children included (35% of all invited), their results correspond to those in current literature and have implications for professionals and parents of preterm children. The study gives insight into the presence of FSD among Slovenian VLBW preterm children, the factors that may predict these disorders, and the long-term consequences they have on the preterm children. The effect of FSD on the child’s growth and development is very important information.

The manuscript is clear, relevant for the field and presented in a well-structured manner.

The cited references mostly recent publications and relevant. The manuscript does not include an excessive number of self-citations.

The manuscript is scientifically sound and the manuscript’s results are reproducible based on the details given in the methods section.

The tables are appropriate and they properly show the data. They are easy to interpret and understand.

The data is interpreted appropriately and consistently throughout the manuscript.

The conclusions are consistent with the evidence and arguments presented.

A questionnaire and a signed informed consent were filled by the parents, but authors did not clearly stated that the study was approved by any ethical board.

The authors should have presented the data, if there is any, on SLP support during swallowing habilitation after discharge from hospital.

Author Response

Dear reviewer,

we are grateful for your review and valuable comments. We tried to follow your suggestions.

Point 1.

A questionnaire and a signed informed consent were filled by the parents, but authors did not clearly state that the study was approved by any ethical board.

The data on ethical board approval have been added in Materials and Methods section (lines 96-99):

The study was carried out in accordance with The Code of Ethics of the World Medical Association (Declaration of Helsinki) and approved by National Medical Ethics Committee of the Republic of Slovenia (Protocol No. 0120-112/2018/4).

Point 2.

The authors should have presented the data, if there is any, on SLP support during swallowing habilitation after discharge from hospital.

We have collected the information on children receiving SLP support after discharge from the hospital and have inserted the data in the paper - Results (lines 242-247):

Thirteen children with FSD were included in speech and language therapy, namely nine children only for language and articulation problems, and five children for FSD. These five children were referred to a tertiary hospital specialized in rehabilitation of FSD at a mean age of 18 months. Four children were included in therapy for sensory-motor feeding problems, while the parents of one child were counseled about the child’s delayed feeding skills.

Discussion (lines 375-379):

Only 13 children with FSD were included in speech therapy, but only five were actually treated for FSD. The reason for such a small proportion of children who received professional help could be poor access to FSD rehabilitation for children in our country. There are only several institutions with a limited number of properly trained speech pathologists who deal with this problem.

English language and style have been checked again by English native speaker.

We do hope that our corrections will meet your expectations.

Thank you again for the comments and encouraging evaluation of our study.

Sincerely yours,

Lilijana Kornhauser Cerar

Reviewer 2 Report

This is an original article about about feeding and swallowing disoreders in former VLBW infants.

However, there are some issues in Methodology that need to be clarified.  The parameter of birth length is analyzed as mean/SD of absolute number. It could be more appropriate to use it as mean/SD of percentile for gestational age (as it was done for birth weight). Also, there is an interesting remark of influence of obtaining proper length relative to weight (probably ponderal index?) of final outcome (lines 265-266). Therefor, it maybe reasonable to use this parameter as well in analysis as a risk factor.

To my opinion, using of these parameter would strengthen the key argument of this paper that lower birth length is a risk factors for FSD in VLBW infants.

Author Response

Dear reviewer,

we are grateful for your review and valuable comments. We tried to follow your suggestions.

Point 1.

The parameter of birth length is analyzed as mean/SD of absolute number. It could be more appropriate to use it as mean/SD of percentile for gestational age (as it was done for birth weight). Also, there is an interesting remark of influence of obtaining proper length relative to weight (probably ponderal index?) of final outcome (lines 265-266). Therefore, it may be reasonable to use this parameter as well in analysis as a risk factor.

According to your suggestion, we have supplemented the input data with markers of intrauterine growth restriction (percentile of birth length - birth length below 10th centile, ponderal index) and conducted a new statistical analysis. Both indicators of inappropriate fetal growth have not been confirmed as a significant factor influencing FSD in the VLBW infants.

The variables have been included in Table 1 / 2 and explained in lines 205-209:

On the other hand, having the birth length below the 10th centile (“short for gestational age”) was not a risk factor for FSD and there was no significant difference between the two groups with respect to ponderal index (a marker to assess body proportionality with the aim to distinguish symmetric from asymmetric growth restriction at birth, calculated by the formula: weight (g) / (length in cm)3 x 100).

Hence, analysis of our data confirmed gestational age of less than 28 weeks, birth weight equal or less than 1000 grams, and birth length below 33 cm as risk factors for FSD. Initial and additional analysis did not suggest that intrauterine growth restriction with weight or length below 10th centile and low ponderal index could predict FSD.

English language and style have been checked again by English native speaker.

Once again, we thank you for your justified comments. We do hope that our additions will meet your expectations.

Sincerely yours,

Lilijana Kornhauser Cerar